# Food transfers, electronic food vouchers and child nutritional status among Rohingya children living in Bangladesh

**John Hoddinott**[1,2]*, **Paul Dorosh**[2], **Mateusz Filipski**[2,3], **Gracie Rosenbach**[2], **Ernesto Tiburcio**[2,4]

**1** Cornell University, Ithaca, New York, United States of America, **2** International Food Policy Research Institute (IFPRI), Washington, DC, United States of America, **3** University of Georgia, Athens GA, United States of America, **4** Tufts University, Medford MA, United States of America

* jfh246@cornell.edu

## Abstract

### Objective

To examine associations between receipt of an electronic food voucher (e-voucher) compared to food rations on the nutritional status of Rohingya children living in refugee camps in Bangladesh.

### Methods

This is an associational study using cross-sectional data. We measured heights and weights of 523 children aged between 6 and 23 months in households receiving either a food ration consisting of rice, pulses, vegetable oil (362 children) or an e-voucher (161 children) that could be used to purchase 19 different foods. Data were also collected on the characteristics of their mothers and the households in which they lived, including household demographics, consumption and expenditure, coping strategies, livelihoods and income profiles, and access to assistance. Associations between measures of anthropometric status (height-for-age z scores, stunting, weight-for-height z scores, wasting, weight-for-age z scores and mid-upper arm circumference) and household receipt of the e-voucher were estimated using ordinary least squares regressions. Control variables included child, maternal, household and locality characteristics. The study received ethical approval from the Institutional Review Board of the International Food Policy Research Institute, Washington DC.

### Results

Household receipt of an e-voucher was associated with improved linear growth in children. This association is robust to the inclusion of maternal, household and location characteristics. The magnitude of the association is 0.38 SD (CI: 0.01, 0.74), and statistically significant at the five percent level. We cannot reject the null hypothesis that these associations differ by child sex. Receipt of an e-voucher is not associated with stunting when a full set of control variables are included. There is no association between receipt of e-vouchers and weight-

**Data Availability Statement:** Data are available at https://doi.org/10.7910/DVN/5BAN6C.

**Funding:** The authors received no specific funding for this work.

**Competing interests:** The authors have declared that no competing interests exist.

for-length, weight-for-age or mid-upper arm circumference. We cannot reject the null hypothesis that these associations differ by child sex.

## Conclusions

In a humanitarian assistance setting, Rohingya refugee camps in Bangladesh, household receipt of an electronic food voucher instead of a food ration is associated with improvements in the linear growth of children between 6 and 23 months but not in measures of acute undernutrition or other anthropometric outcomes. Our associational evidence indicates that transitioning from food rations to electronic food vouchers does not adversely affect child nutritional status.

## Introduction

Globally, the number of forcibly displaced persons now exceeds 70 million [1]. Considerable efforts, human and financial resources are devoted to providing humanitarian assistance to these persons. Until recently, nearly all this assistance, 94%, has been provided in-kind, but there is interest in shifting this to cash or vouchers [2]. These are seen to be less costly to deliver, allow beneficiaries greater choice in using transfers that they receive while enhancing the transparency surrounding how much of humanitarian assistance actually reaches its intended beneficiaries [2].

However, unlike evaluations of cash, voucher and food transfers in non-humanitarian settings (see Gentilini [3] for a recent review and Ahmed, Hoddinott and Roy for evidence on this topic in Bangladesh [4]) relatively little is known about the impacts of such a shift on children's chronic and acute nutritional status in humanitarian settings. Seal, Dolan Trenouth [5] summarize the findings of the REFANI project, that implemented in three countries to strengthen the evidence base on the nutritional impact and cost-effectiveness of cash and voucher transfers to populations affected by humanitarian emergencies. In Somalia, relative to a control group, REFANI found that providing cash had no impact on global acute malnutrition (GAM), height-for-age z scores (HAZ) or stunting. The REFANI Pakistan study assessed the impact of providing cash for six months, a "double cash" transfer and a fresh food voucher worth relative to a control group that received no payments. Relative to the control group, the double cash and the fresh food voucher treatment arms improved and the double cash treatment arm reduced wasting. Relative to control, cash, the double cash and fresh food voucher arms improved HAZ and reduced stunting. The authors did not report if these impacts differed across treatment arms. In Niger, the REFANI study found no evidence that providing cash to poor household for six months rather than four had no impacts on weight-for-height z scores (WHZ), GAM, HAZ, or stunting. A different study, also situated in Niger, assessed the impact of providing cash, finding that relative to a control group, children in the cash treatment arm saw greater improvements in WHZ [6]. Finally, in the Democratic Republic of Congo, Grellety et al [7] find that children in households receiving cash transfers recover from severe acute malnutrition faster than children in control households who did not receive a cash transfer.

While these studies provide new knowledge on the impact of transfers on children's nutritional status–most notably the findings from the Pakistan and one of the Niger studies that show that cash transfers can improve some dimensions of nutritional status—they do not tell us whether shifting transfer modalities (ie from food to cash or food to vouchers) will affect

the acute or chronic nutritional status of pre-school children living in areas experiencing humanitarian crises. Consistent with this observation, a recent research agenda setting exercise indicated that there was little evidence on the comparative efficiency or effectiveness of cash or vouchers (what the authors refer to as cash transfer programming) on health and nutrition outcomes in humanitarian settings [8].

This paper seeks to help fill this evidence gap. We examine associations between two forms of unconditional food assistance–a food ration and an electronic voucher which can be used to purchase a wide range of foods—and the anthropometric status of Rohingya (Forcibly Displaced Myanmar Nationals or FDMN) children living in camps for FDMN in Bangladesh. Our motivation is twofold: (1) as noted above, relatively little is known about the comparative effects of food and voucher payments on children's nutritional status; and (2) the World Food Programme, who oversee food assistance to the Rohingya, are anticipating increasing the number of beneficiary households who receive these vouchers and so it would be helpful to know what the consequences of such a shift would be for children's nutritional status. This is of particular importance given that, in 2017, more than 40 percent of Rohingya children aged 6 months to five years were stunted and between 14 and 24 percent exhibited GAM [9].

## Setting and methods

### Setting and sample

The Rohingya are a Muslim minority ethnic group from the western part of Myanmar's Rakhine State. They have suffered persecution for several decades, lack citizenship rights, have poor access to public services, and limited freedom of movement [10]. Since 1978, members of this community have been periodically forced to flee to the safety of Bangladesh; by August 2017, 213,000 Rohingya were living in two registered camps (Kutupalong and Nayapara) in the Cox's Bazar District of southeastern Bangladesh. Renewed, intense violence forced a mass exodus out of Myanmar with more than 671,000 Rohingya individuals fleeing to Bangladesh between August and October 2017. These new arrivals were placed in makeshift camps hastily constructed on land provided by the Government of Bangladesh. By mid-2019 there were an estimated 911,000 Rohingya living in Bangladesh [1], [11]. Only a small fraction (about 50 thousand people who arrived in the 1990s) have been given refugee status; consequently, the Rohingya in Bangladesh are formally referred to as Forcibly Displaced Myanmar Nationals. They are nearly wholly reliant on assistance provided by the international community for food, shelter and medical needs. There is little variation in the quality of housing of these households with most living in dwellings constructed out of bamboo or plastic/polythene with mud floors. Apart from mobile phones and cooking utensils, ownership of consumer durables is limited; for example, among newer migrants, fewer than 20 percent own a bed, table or chair. These are poor households.

In October and November 2018, we conducted a cross-sectional survey of FDMN in collaboration with the Bangladesh Institute of Development Studies and the non-governmental organization Action Against Hunger. The purpose of the survey was to document the food security and nutrition status of the Rohingya, to understand how they generated income, assess the extent to which they relied on food coping strategies and to review the role played by food assistance efforts provided by international donors in improving food security and nutrition. It built on previous survey instruments fielded in these localities; modules not previously implemented were pre-tested before the questionnaire was fielded. The survey included both Rohingya who had arrived before August 2017—the older wave of migrants—and those arriving after the mass expulsion in September and October 2017—the newer wave of migrants. The Rohingya in the older migration waves were selected through simple random sampling in

the Nayapara Registered Camp. The Rohingya in the most recent migration wave were sampled using two-stage clustered random sampling, using the blocks (sub-camps) within each camp (strata) as the clusters. The final sample consists of 1,308 Rohingya households from the newer wave, 781 Rohingya households from the older waves. The survey instrument covered household demographics, consumption and expenditure, coping strategies, nutrition outcomes, livelihoods and income profiles, and access to assistance. These data were collected by enumerators trained and employed by the Bangladesh Institute of Development Studies and Action Against Hunger. Collection of the anthropometric data drew on the Standardized Monitoring and Assessment of Relief and Transitions (SMART) methodology used by international organizations and humanitarian practitioners. SMART includes methods for standardizing anthropometric measurements as well as software to check data quality and flag problems [12].

For this paper, we undertook a sub-analysis of these data. Specifically, we focus on those children in our sampled households who were aged between 6 and 23 months at the time of the survey. Children in this age range are within the "1000 days window" critical for child nutrition. They are no longer being exclusively breastfed; for children in this age range, the availability of adequate quantities of a range of nutritious foods is critical for their health and development. In our sample, there are 523 children in this age range. 362 children live in households receiving the food ration; 161 children live in households receiving the e-voucher. An ex-post power calculation showed that this sample was large enough to detect a: 0.35SD difference in height-for-age z score between children living in households receiving food rations and e-vouchers; and a 0.25SD difference in weight-for-height z scores between children living in households receiving food rations and e-vouchers. We received permission from the Government of Bangladesh, Office of the Refugee Relief and Repatriation Commissioner, Cox's Bazar to conduct this survey. It received ethical approval from the Institutional Review Board of the International Food Policy Research Institute, Washington DC. Because of high levels of illiteracy in the localities where this study took place, oral consent to participate in the survey was received from participants and this oral consent was witnessed and formally recorded. As this survey included measuring the heights and weights of children, we obtained oral consent from their parents and guardians before taking these measurements. This consent process was approved by the Institutional Review Board of the International Food Policy Research Institute.

## Access to food assistance

Food assistance to FDMN is provided by the United Nations World Food Programme (WFP). Access to this assistance is virtually universal in the full sample. Consistent with WFP administrative reports [11], fewer than one percent of our total sample stated that they were not receiving assistance when the survey took place. Households not receiving assistance are excluded from our analysis. Food assistance takes two forms. Households selected for the General Food Distribution (GFD) receive rice, lentils and micronutrient fortified cooking oil at one of 21 designated distribution points. The size of the ration and the frequency of payment depend on household size. At the time of the survey, households with 1–3 members received 30 kg of rice, 9 kg of lentils and 3 liters of cooking oil with these payments made monthly. Households with 4–7 members received 30 kg of rice, 9 kg of lentils and 3 liters of cooking oil with these payments made twice per month. Households with more than 8 members received 60 kg of rice, 13.5 kg of lentils and 6 liters of cooking oil with these payments made twice per month [13]. Re-sale of the GFD is relatively uncommon. 76% of households that received some WFP

assistance reported not selling it. Only 15% said they sold it for food aid and 12% exchanged or bartered it for other commodities.

The second form of assistance is an electronic food voucher or e-voucher. Households receiving the e-voucher received a payment of 760–800 Taka (approximately 9 USD) per household member per month). The holder of the e-voucher is the senior woman in the household to whom payments are made [14]. The e-voucher itself resembles a debit card with an electronic chip. The chip enables the monthly payment to be made remotely without a need for beneficiaries to travel to or line up at a pay point. The card can only be used at designated shops within the camps that are equipped with point-of-sale machines that can read these cards, and only for 19 designated foods. These foods include the items found in the GFD–rice, lentils and cooking oil–but also fruit, vegetables, dried fish, eggs as well as various spices. Prices for these foods are negotiated by WFP and are fixed at levels which equalized the value of the e-voucher and the GFD.

At the time of the survey, across all households interviewed (ie the 1,308 Rohingya households from the newer wave of refugees and 781 Rohingya from the older waves of refugees), 62%, received GFD, 34% received e-vouchers and 4% received both. Households reporting that they received both forms of assistance are excluded from our analysis. As part of our research for this study, in May 2019 we met the Emergency Coordinator for the World Food Programme Rohingya Refugee Response and his staff. He indicated that these proportions reflected the amount of cash and of food that WFP had available to distribute. He noted that there was no formal process or decision rule for the allocation of beneficiaries to GFD or e-vouchers nor was there written documentation on these allocations. He and his staff described the process of selecting areas where e-vouchers would be used in the following way. The use of e-vouchers required new shops to be constructed and shops require land. WFP was not permitted to purchase this land. Instead, the camps were demarcated into five catchment areas and within these, the Government of Bangladesh provided land (where available) at certain places within each for shops to be located. There was no formal process or criteria for the decision to provide land (and build a shop) in any given place and WFP had no choice regarding where these shops were sited; maps provided to us by WFP show that most of these shops are co-located in places where the GFD is given out. Once this decision was made, a certain number of households in the proximity of the shop were enrolled into e-vouchers. Beneficiaries themselves had no choice as to whether they would receive GFD or e-vouchers. Access to e-vouchers was not randomized and thus our study is associational. However, WFP staff indicated that access to e-vouchers was not linked to specific child or household characteristics.

## Outcome measures

Studies of the impact of cash or food interventions in humanitarian settings focus on short-term measures of nutritional status such as weight-for-height and acute undernutrition, wasting [5], [6], [7]. But because the Rohingya are expected to remain in these refugee camps for a considerable length of time, it is also of interest to see how these different transfer modalities affect child growth. For these reasons, using the data collected on heights and weights of children between 6 and 23 months along with information on child age, we calculated a continuous measure of attained linear growth, length/height-for-age z scores (HAZ) using the WHO growth standards [10]. Using this measure, we constructed a dichotomous measure of linear growth, stunting. A child is considered stunted if she had a HAZ $< -2.0$. We constructed a continuous measure of thinness, weight-for-height z scores (WHZ) also using WHO growth standards. Using this measure, we constructed a dichotomous measure of acute undernutrition, wasting. This equaled one if the child had a WHZ $< -2.0$. Note, however, that because

WHZ includes both weight and height in its construction, children who experience changes in both weights and heights may not see changes in WHZ. For this reason, we also calculated weight-for-age z scores (WHZ) using the WHO growth standards [15] and we measured, and report, estimates where the outcome measure is mid-upper arm circumference (MUAC).

### Statistical analysis

Statistical analyses were conducted in STATA 16 (StatCorp LP). Descriptive statistics, means and standard deviations, were calculated for outcome and control variables. We constructed density functions to compare the distributions of our continuous outcomes, HAZ and WHZ, according to whether the child was in a household receiving a GFD or an e-voucher.

We estimate associations between household receipt of an e-voucher and child nutritional status. These associations are assessed relative to the base category, household receipt of the GFD. In our base specification, we control for frequency of payment receipt and child age and sex. To assess the robustness of our estimated associations and informed by the UNICEF conceptual framework [16] for understanding the correlates of child nutritional status we sequentially add: maternal characteristics (age and whether the mother has any formal schooling); household characteristics (dummy variables for whether the household is a new migrant household—namely, arriving after August 2017—, whether the household is female-headed, whether the household head has any formal schooling, an asset index calculated using principal components analysis based on ownership of consumer durables and productive assets and a dummy variable indicating if the dwelling observed to exhibit any kind of damage to the walls or roof); and locality characteristics (whether the household resided in a registered refugee camp; log distance to the closest food distribution point; log distance to the closest primary health clinic).

We use ordinary least squares regressions for continuous (HAZ, WHZ, WAZ and MUAC) outcomes as well as dichotomous outcomes (stunting, wasting); the latter are estimated as linear probability models. As a robustness check, we estimate probit models for our dichotomous outcomes. We disaggregate our results to assess whether these associations differ by child sex. We also estimate a model where in addition to our base specification and child, maternal, household and location characteristics, we include an interaction term between child sex and receipt of an e-voucher to assess whether e-voucher receipt has differential impacts by sex. Standard errors are robust to heteroscedasticity of unknown form; as many of our clusters have only a few children, we do not account for clustering at the sub-camp level.

## Results

Our sample for the nutrition analysis is nearly evenly split between girls (52 percent) and boys (Table 1). Mean age is 15.5 months; 11 percent of children are not offspring of the household head. 21 percent of mothers have any formal education. Just under 70 percent of our sample consists of children who reside in households who arrived in Bangladesh after August 2017. 17 percent, of households are female headed and 29 percent of all household heads have any formal schooling. Twenty percent live in the Nayapara Registered Camp. Mean distance to the nearest food distribution point is just over 1km. Health clinics are, over average, 6km away.

Table 1 also disaggregates these characteristics by household receipt of the food ration or the e-voucher. Children in households receiving the food ration had poorer HAZ and were more likely to be stunted, 36 percent compared to 27 percent of children in households receiving the e-voucher. Measures of acute undernutrition, WHZ and wasting, are comparable across the two groups as are WAZ and MUAC. Some characteristics are similar across the

**Table 1. Descriptive statistics.**

| | Unit | All Children | | | Children in households receiving food ration | | | Children in households receiving e-voucher | | | P value for t test on difference in means |
|---|---|---|---|---|---|---|---|---|---|---|---|
| | | Mean | Standard Deviation | Sample size | Mean | Standard Deviation | Sample size | Mean | Standard Deviation | Sample size | |
| Outcomes | | | | | | | | | | | |
| Length for age | Z score | -1.55 | 1.51 | 523 | -1.63 | 1.57 | 362 | -1.35 | 1.35 | 161 | 0.05 |
| Stunted | Percent | 33.4 | 47.2 | 523 | 36.1 | 48.1 | 362 | 27.3 | 44.7 | 161 | 0.05 |
| Weight-for-length | Z score | -1.09 | 0.93 | 516 | -1.07 | 0.94 | 360 | -1.11 | 0.92 | 156 | 0.84 |
| Wasted | Percent | 15.8 | 36.6 | 516 | 16.1 | 36.8 | 360 | 15.3 | 36.2 | 156 | 0.64 |
| Weight-for-age | Z score | -1.61 | 1.08 | 517 | -1.64 | 1.08 | 360 | -1.54 | 1.08 | 156 | 0.36 |
| MUAC | mm | 137.14 | 9.12 | 523 | 137.17 | 9.24 | 362 | 137.07 | 8.86 | 161 | 0.90 |
| Access to assistance | | | | | | | | | | | |
| Household receives e-voucher | Percent | 30.7 | 46.2 | 523 | - | - | | - | - | | - |
| Household receives payment every 15 days | Percent | 56.9 | 49.5 | 523 | 77.3 | 41.9 | 362 | 11.1 | 31.6 | 161 | <0.01 |
| Child characteristics | | | | | | | | | | | |
| Female | Percent | 52.2 | 50.0 | 523 | 54.1 | 49.9 | 362 | 47.8 | 50.1 | 161 | 0.18 |
| Age | Months | 15.5 | 4.7 | 523 | 15.5 | 4.5 | 362 | 15.5 | 5.0 | 161 | 0.89 |
| Child not offspring of household head | Percent | 11.8 | 32.3 | 523 | 7.4 | 26.3 | 362 | 21.7 | 41.3 | 161 | <0.01 |
| Maternal characteristics | | | | | | | | | | | |
| Age | Years | 29.2 | 9.3 | 523 | 28.3 | 8.9 | 362 | 31.1 | 10.0 | 161 | <0.01 |
| Any formal education | Percent | 21.2 | 40.9 | 523 | 21.2 | 40.9 | 362 | 21.1 | 40.9 | 161 | 0.96 |
| Household characteristics | | | | | | | | | | | |
| New migrant household (arrived after August 2017) | Percent | 69.9 | 45.8 | 523 | 84.5 | 36.2 | 362 | 37.2 | 48.5 | 161 | <0.01 |
| Female-headed | Percent | 17.4 | 37.9 | 523 | 14.0 | 34.8 | 362 | 24.8 | 43.3 | 161 | <0.01 |
| Head has any formal education | Percent | 28.8 | 45.3 | 523 | 28.7 | 45.3 | 362 | 29.1 | 45.6 | 161 | 0.91 |
| Wealth index | | -0.28 | 0.72 | 523 | -0.38 | 0.59 | 362 | -0.05 | 0.92 | 161 | <0.01 |
| Dwelling shows damage | Percent | 65.9 | 47.4 | 523 | 66.8 | 47.1 | 362 | 63.9 | 48.1 | 161 | 0.52 |
| Location characteristics | | | | | | | | | | | |
| Household resides in a registered refugee camp | Percent | 20.4 | 40.38 | 523 | 6.9 | 25.3 | 362 | 50.9 | 50.1 | 161 | <0.01 |
| Distance to nearest Primary Health Clinic | Meters | 6,050 | 7,144 | 523 | 3,384 | 5,789 | 362 | 12,044 | 6,218 | 161 | <0.01 |
| Distance to nearest Food Distribution Post | Meters | 1,028 | 1,115 | 523 | 887.7 | 1,200 | 362 | 1,342 | 815.4 | 161 | <0.01 |

Source, Household questionnaire.

food ration and e-voucher groups: child age, maternal age and education, education of the head, whether the dwelling shows any damage. But other characteristics do differ. Relative to children in households receiving the food ration, children in households receiving e-vouchers are: (i) less likely to be female and to have arrived after August 2017; (ii) more likely to not be the biological offspring of the household head, to live in a registered camp and to live farther away from a health clinic or a food distribution point.

Fig 1 shows the density functions for length-for-age z scores, disaggregated by whether the child was in a household receiving assistance through the GFD or through the e-voucher. For both, the mass of the distributions lies to the left of zero, indicating that this is a poorly

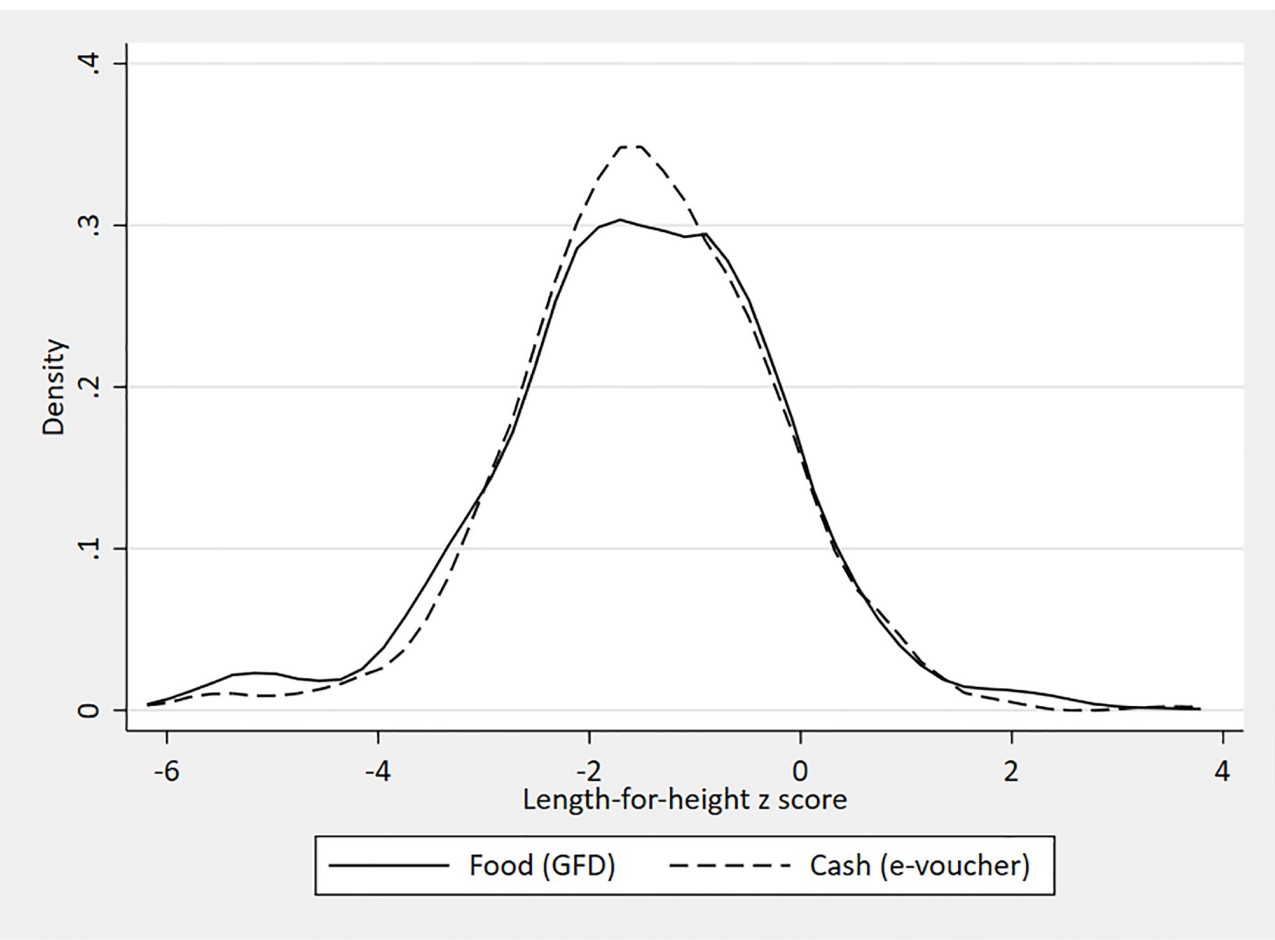

**Fig 1. Kernel density plot for length-for-age z score, by payment modality.**

nourished sample of children. At lower z score values, the distribution for children in households receiving e-vouchers lies to the right of that for children in GFD households. Fig 2 shows the density functions for weight-for-length z scores, again disaggregated by payment type. Again, the distributions lie to the left of zero, indicating the acute undernutrition is a concern in this population but unlike Fig 1, there is little difference in the distributions as disaggregated by payment type. Consistent with Figs 1 and 2, Table 1 provides further evidence of poor nutritional status. Across the full sample, mean HAZ is -1.55 and 33.4 percent of children are stunted. Mean WHZ is -1.09 and 15.9 percent of children are wasted.

Table 2 reports associations between receipt of the e-voucher and linear growth (columns 1–4) Column (1) shows a positive association, 0.46 SD, between household e-voucher receipt and HAZ; this association is statistically significant at the 1 percent level. Adding maternal characteristics (column 2), household characteristics (column 3) and location characteristics (column 4) reduces the magnitude of the association but even in the presence of all the controls (column 4), the association is 0.38SD and statistically significant at the 5 percent level.

Columns 5–8 of Table 2 replicate this analysis using stunting as the outcome variable. In our basic model (column 5), the e-voucher is associated with a 11 percentage point reduction in the likelihood that the child is stunted; this association is statistically significant at the 10

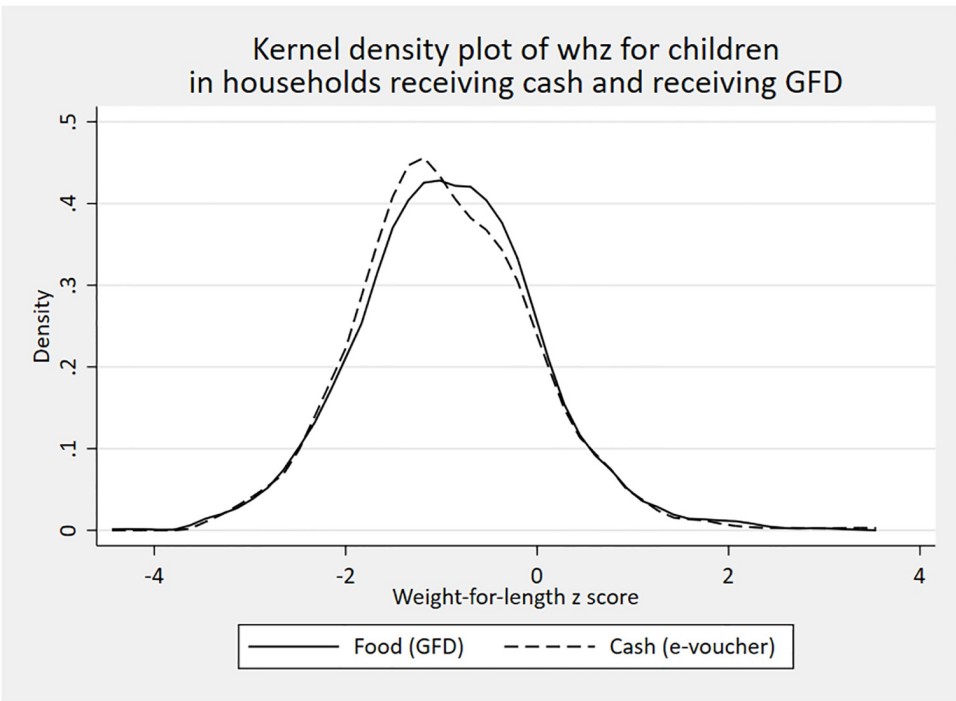

**Fig 2. Kernel density plot for weight- for-height z score, by payment modality.**

percent level. Adding controls reduces the magnitude of this association (eight percentage points when all controls are included) and it is no longer statistically significant.

Table 3 reports associations between e-vouchers and WHZ (columns 1–4) and a measure of acute undernutrition, wasting (columns 5–8). The associations between e-vouchers and WHZ and between e-vouchers and wasting are not statistically significant.

Table 4 reports associations between e-vouchers and WAZ (column 1) and MUAC (column 2) with all controls included. The associations between e-vouchers and WAZ and between e-vouchers and MUAC are not statistically significant. (Note, for brevity, we do not report specifications with fewer controls; these show no statistically significant associations).

Table 5 disaggregates our HAZ and stunting results by child sex, controlling for frequency of payment, child, maternal, household and location characteristics. Column (1) shows for girls, a positive association, 0.45 SD, between household e-voucher receipt and HAZ; this association is statistically significant at the 5 percent level. For boys, the association between HAZ and household receipt of an e-voucher is smaller and not statistically significant (column 3). However, we also estimate a model where in addition to our base specification and child, maternal, household and location characteristics, we include an interaction term between child sex and receipt of an e-voucher. This interaction term is not statistically significant. There does not appear to be child sex differences in the associations between household receipt of an e-voucher and stunting, weight-for-height or wasting.

## Discussion

To the best of our knowledge, relatively little is known about the impact of shifting from in-kind transfers, such as food, to the provision of food vouchers on the nutritional status of children in humanitarian settings. We attempt to remedy this evidence gap by examining

**Table 2. Associations between receipt of e-voucher and chronic undernutrition.**

| | Length for age z score | | | | Stunted | | | |
|---|---|---|---|---|---|---|---|---|
| | **(1)** | **(2)** | **(3)** | **(4)** | **(5)** | **(6)** | **(7)** | **(8)** |
| Access to assistance | | | | | | | | |
| Household receives e-voucher | 0.46*** | 0.45** | 0.39** | 0.38** | -0.11* | -0.10* | -0.09 | -0.08 |
| | (0.18) | (0.18) | (0.18) | (0.19) | (0.06) | (0.06) | (0.06) | (0.06) |
| Household receives payment every 15 days | 0.22 | 0.21 | 0.26 | 0.37* | -0.00 | 0.00 | -0.02 | -0.05 |
| | (0.17) | (0.17) | (0.18) | (0.20) | (0.05) | (0.05) | (0.06) | (0.06) |
| Child characteristics | | | | | | | | |
| Female | 0.03 | 0.02 | 0.01 | -0.02 | -0.02 | -0.02 | -0.02 | -0.01 |
| | (0.13) | (0.13) | (0.13) | (0.13) | (0.04) | (0.04) | (0.04) | (0.04) |
| Age, child | -0.08*** | -0.08*** | -0.08*** | -0.08*** | 0.02*** | 0.01*** | 0.01*** | 0.02*** |
| | (0.01) | (0.01) | (0.01) | (0.01) | (0.00) | (0.00) | (0.00) | (0.00) |
| Child not offspring of household head | -0.21 | -0.25 | -0.28 | -0.35 | 0.10 | 0.12 | 0.12 | 0.15 |
| | (0.19) | (0.30) | (0.30) | (0.30) | (0.07) | (0.09) | (0.09) | (0.09) |
| Maternal characteristics | | | | | | | | |
| Age, mother | | 0.00 | 0.00 | 0.00 | | -0.00 | -0.00 | -0.00 |
| | | (0.01) | (0.01) | (0.01) | | (0.00) | (0.00) | (0.00) |
| Mother has any formal education | | 0.22 | 0.09 | 0.09 | | -0.08* | -0.06 | -0.05 |
| | | (0.15) | (0.18) | (0.18) | | (0.05) | (0.06) | (0.06) |
| Household characteristics | | | | | | | | |
| Newer migrant household | | | 0.21 | 0.01 | | | -0.07 | 0.00 |
| | | | (0.16) | (0.21) | | | (0.06) | (0.07) |
| Female headed household | | | -0.11 | -0.12 | | | -0.02 | -0.01 |
| | | | (0.19) | (0.19) | | | (0.06) | (0.06) |
| Head has any formal education | | | 0.14 | 0.14 | | | -0.06 | -0.05 |
| | | | (0.17) | (0.17) | | | (0.05) | (0.05) |
| Wealth index | | | -0.00 | -0.03 | | | 0.02 | 0.03 |
| | | | (0.07) | (0.07) | | | (0.02) | (0.02) |
| Dwelling shows damage | | | -0.27* | -0.29* | | | 0.01 | 0.01 |
| | | | (0.15) | (0.16) | | | (0.05) | (0.05) |
| Location characteristics | | | | | | | | |
| Registered Camps | | | | 0.42* | | | | -0.16* |
| | | | | (0.26) | | | | (0.09) |
| Log distance to Primary Health Clinic | | | | -0.00 | | | | 0.00 |
| | | | | (0.00) | | | | (0.00) |
| Log distance to Food Distribution Post | | | | 0.00** | | | | -0.00 |
| | | | | (0.00) | | | | (0.00) |
| Observations | 523 | 523 | 523 | 523 | 523 | 523 | 523 | 523 |
| R-squared | 0.07 | 0.08 | 0.09 | 0.10 | 0.03 | 0.04 | 0.04 | 0.05 |

Robust standard errors in parentheses.

* significant at the 10% level;

** significant at the 5% level;

*** significant at the 1% level. Sample size is 523. 362 children live in households receiving the food ration; 161 children live in households receiving the e-voucher.

associations between forms of food assistance–a food ration and an electronic voucher which can be used to purchase a wide range of foods—and the anthropometric status of Rohingya (Forcibly Displaced Myanmar Nationals) children living in refugee camps in Bangladesh. Controlling for child, maternal, household and location characteristics, we find that receipt of an

**Table 3. Associations between receipt of e-voucher and acute undernutrition.**

| | Weight-for-length z-score | | | | Wasted | | | |
|---|---|---|---|---|---|---|---|---|
| | (1) | (2) | (3) | (4) | (5) | (6) | (7) | (8) |
| Access to assistance | | | | | | | | |
| Household receives e-voucher | -0.07 | -0.07 | -0.08 | -0.02 | -0.00 | -0.01 | -0.02 | -0.02 |
| | (0.11) | (0.12) | (0.12) | (0.12) | (0.05) | (0.05) | (0.05) | (0.05) |
| Household receives payment every 15 days | 0.01 | 0.02 | 0.02 | 0.06 | -0.00 | -0.02 | -0.01 | -0.04 |
| | (0.10) | (0.10) | (0.11) | (0.12) | (0.05) | (0.05) | (0.05) | (0.05) |
| Child characteristics | | | | | | | | |
| Female | -0.02 | -0.02 | -0.02 | -0.03 | 0.01 | 0.01 | 0.01 | 0.01 |
| | (0.08) | (0.08) | (0.08) | (0.08) | (0.03) | (0.03) | (0.03) | (0.03) |
| Age, child | 0.01 | 0.01 | 0.01 | 0.01 | -0.00 | -0.00 | -0.00 | 0.00 |
| | (0.01) | (0.01) | (0.01) | (0.01) | (0.00) | (0.00) | (0.00) | (0.00) |
| Child not offspring of household head | 0.23* | 0.26 | 0.24 | 0.20 | -0.05 | -0.15** | -0.14** | -0.12* |
| | (0.12) | (0.17) | (0.17) | (0.18) | (0.05) | (0.06) | (0.07) | (0.07) |
| Maternal characteristics | | | | | | | | |
| Age, mother | | -0.00 | -0.00 | -0.00 | | 0.00** | 0.00** | 0.01** |
| | | (0.01) | (0.01) | (0.01) | | (0.00) | (0.00) | (0.00) |
| Mother has any formal education | | -0.08 | -0.15 | -0.16 | | -0.04 | -0.03 | -0.02 |
| | | (0.10) | (0.11) | (0.11) | | (0.04) | (0.04) | (0.04) |
| Household characteristics | | | | | | | | |
| Newer migrant household | | | 0.00 | -0.09 | | | 0.04 | 0.10 |
| | | | (0.11) | (0.14) | | | (0.05) | (0.06) |
| Female headed household | | | 0.03 | 0.03 | | | -0.05 | -0.04 |
| | | | (0.11) | (0.11) | | | (0.04) | (0.05) |
| Head has any formal education | | | 0.09 | 0.09 | | | -0.01 | -0.01 |
| | | | (0.10) | (0.10) | | | (0.04) | (0.04) |
| Wealth index | | | 0.02 | 0.02 | | | -0.01 | -0.01 |
| | | | (0.04) | (0.04) | | | (0.02) | (0.02) |
| Dwelling shows damage | | | -0.10 | -0.13 | | | 0.03 | 0.03 |
| | | | (0.09) | (0.10) | | | (0.04) | (0.04) |
| Location characteristics | | | | | | | | |
| Registered Camps | | | | 0.32* | | | | -0.15** |
| | | | | (0.17) | | | | (0.07) |
| Log distance to Primary Health Clinic | | | | -0.00* | | | | 0.00 |
| | | | | (0.00) | | | | (0.00) |
| Log distance to Food Distribution Post | | | | 0.00 | | | | -0.00 |
| | | | | (0.00) | | | | (0.00) |
| Observations | 516 | 516 | 516 | 516 | 516 | 516 | 516 | 516 |
| R-squared | 0.01 | 0.01 | 0.02 | 0.02 | 0.00 | 0.00 | 0.01 | 0.02 |

Robust standard errors in parentheses.

* significant at the 10% level;

** significant at the 5% level;

*** significant at the 1% level. Sample size is 516. 360 children live in households receiving the food ration; 156 children live in households receiving the e-voucher.

e-voucher is associated with an increase in HAZ of 0.38SD. This association is statistically significant at the 5 percent level. When we include a full set of control variables, we do not find an association between receipt of e-vouchers and stunting. Fig 1 suggests that the association of e-vouchers and HAZ is concentrated on children with very low HAZ scores, below the -2

**Table 4. Associations between receipt of e-voucher and weight-for-age and MUAC.**

| | Weight-for-age z score | MUAC |
|---|---|---|
| | **(1)** | **(2)** |
| Access to assistance | | |
| Household receives e-voucher | 0.16 | -0.65 |
| | (0.15) | (1.07) |
| Household receives payment every 15 days | 0.25* | 0.46 |
| | (0.15) | (1.11) |
| Child characteristics | | |
| Female | -0.06 | -4.01*** |
| | (0.10) | (0.79) |
| Age, child | -0.03*** | 0.41*** |
| | (0.01) | (0.08) |
| Child not offspring of household head | -0.02 | 1.01 |
| | (0.22) | (1.56) |
| Maternal characteristics | | |
| Age, mother | 0.00 | -0.01 |
| | (0.01) | (0.06) |
| Mother has any formal education | -0.09 | -1.53 |
| | (0.14) | (1.05) |
| Household characteristics | | |
| Newer migrant household | -0.06 | -0.07 |
| | (0.15) | (1.25) |
| Female headed household | -0.05 | -0.17 |
| | (0.13) | (1.01) |
| Head has any formal education | 0.17 | 1.67* |
| | (0.12) | (0.92) |
| Wealth index | -0.02 | -0.06 |
| | (0.05) | (0.38) |
| Dwelling shows damage | -0.24** | -1.66* |
| | (0.11) | (0.91) |
| Location characteristics | | |
| Registered Camps | 0.44** | 1.49 |
| | (0.20) | (1.76) |
| Log distance to Primary Health Clinic | -0.00 | -0.00 |
| | (0.00) | (0.00) |
| Log distance to Food Distribution Post | 0.00** | 0.00 |
| | (0.00) | (0.00) |
| Observations | 517 | 523 |
| R-squared | 0.05 | 0.00 |

Robust standard errors in parentheses.

* significant at the 10% level;

** significant at the 5% level;

*** significant at the 1% level. Sample size is 517 for WAZ and 523 for MUAC.

cut off used to denote stunting. This is consistent with e-vouchers having a positive association with HAZ but not stunting. We cannot reject the null hypothesis that these associations differ by child sex. There are no associations with weight as measured by WHZ or acute undernutrition as measured by wasting. There are no associations with WAZ or MUAC.

**Table 5. Associations between receipt of e-voucher and chronic undernutrition, by sex.**

| | Girls | | Boys | |
|---|---|---|---|---|
| | Length for age z score | Stunted | Length for age z score | Stunted |
| | (1) | (2) | (3) | (4) |
| Household receives e-voucher | 0.54** | -0.14* | 0.23 | 0.01 |
| | (0.24) | (0.08) | (0.31) | (0.08) |
| Observations | 273 | 273 | 250 | 250 |
| R-squared | 0.13 | 0.08 | 0.10 | 0.11 |

See Table 2. Regressions control for frequency of payment, child, maternal, household and location characteristics.

Our study is associational; access to e-vouchers was not randomized. There may be unobserved factors correlated with both receipt of an e-voucher and child anthropometric outcomes; such unobservables will bias our parameter estimates. While we cannot conclusively rule out unobservables, we note that there is no evidence suggesting that e-vouchers were targeted to households with children with atypically good or poor nutritional status. Further, the positive association between e-vouchers and HAZ remains even after controlling for a wide range of child, maternal, household and location characteristics.

Our study has only limited information on why these positive associations exist. We note that women in male headed households receiving the e-voucher were 11 percentage points more likely to decide solely or jointly how to use the food assistance that their households had received compared to women in male headed households receiving the GFD. Households receiving the e-voucher reported being able to make the assistance they received last until the next payment [17]. We also note that there is one published study in a developing country setting, using a randomized control trial design, where treatment arms included a food transfer and a food voucher. This study, fielded in Ecuador, found that both treatments increased the quantity and quality of food consumed by recipient households. However, food transfers led to relatively larger increases in calories consumed while food vouchers led to relatively larger increases in household dietary diversity [18]. Consistent with the Ecuador study, households in our study who received an e-voucher consumed a larger number of food groups than households that received the GFD. If households receiving the e-voucher purchased a wider range of foods then children in this age group may have consumed a more diverse diet, including animal source foods that Semba et al [19] suggest are linked to more rapid linear growth note. We have data on household level consumption but not data on child food consumption; consequently, we are unable to fully investigate this hypothesis. Improved dietary diversity, increased women's control over the transfer and the ability of households to making the transfer last until the next payment are all associated with receipt of the e-voucher. All may have played some role in improving HAZ; our data, however, do not allow us to disentangle their possible effects. We also note that hygiene conditions in these settlements are poor [9] and that poor sanitation is associated with poorer WHZ. Neither e-vouchers or the GFD directly affect hygiene conditions and this may explain why there is no association between e-vouchers and WHZ.

In this setting, our associational evidence indicates that transitioning from food rations to electronic food vouchers does not adversely affect child nutritional status. There is evidence of a positive association between electronic food vouchers and height-for-age z scores. A randomized control trial study, together with collection of data on both nutritional outcomes and the factors that directly affect children's heights and weights would be enormously beneficial in further advancing knowledge on this important issue.

## Supporting information

**S1 File.**
(DOCX)

**S2 File.**
(DOCX)

## Author Contributions

**Conceptualization:** John Hoddinott, Paul Dorosh, Mateusz Filipski, Gracie Rosenbach, Ernesto Tiburcio.

**Data curation:** Gracie Rosenbach, Ernesto Tiburcio.

**Formal analysis:** John Hoddinott, Gracie Rosenbach, Ernesto Tiburcio.

**Writing – original draft:** John Hoddinott, Paul Dorosh, Mateusz Filipski, Gracie Rosenbach, Ernesto Tiburcio.

**Writing – review & editing:** John Hoddinott, Paul Dorosh, Mateusz Filipski, Gracie Rosenbach, Ernesto Tiburcio.

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
