## [Decision Letter · Decision Letter 0]

1 Nov 2019

PONE-D-19-25181

Food transfers, electronic food vouchers and child nutritional status among Rohingya children living in Bangladesh

PLOS ONE

Dear Professor Hoddinott,

Thank you for submitting your manuscript to PLOS ONE. After careful consideration, we feel that it has merit but does not fully meet PLOS ONE’s publication criteria as it currently stands. Therefore, we invite you to submit a revised version of the manuscript that addresses the points raised during the review process.

We would appreciate receiving your revised manuscript by Dec 16 2019 11:59PM. To enhance the reproducibility of your results, we recommend that if applicable you deposit your laboratory protocols in protocols.io, where a protocol can be assigned its own identifier (DOI) such that it can be cited independently in the future. For instructions see: http://journals.plos.org/plosone/s/submission-guidelines#loc-laboratory-protocols

We look forward to receiving your revised manuscript.

Kind regards,

Samson Gebremedhin, PhD

Academic Editor

PLOS ONE

Additional Editor Comments:

Abstract

In the methods sub-section please clearly identify the study design of the study and sample sizes of the two groups being compared.

“The magnitude of the association is large, 0.38 SD” can you please provide the confidence interval for this effect measure?

The sentence “Receipt of an e-voucher is associated with reduced stunting, but this association is imprecisely measured” is not clear.

Introduction

This section looks shallow. Please provide a brief overview of the findings of previous studies on the topic.

Methods

Please describe how the sample size of 523 was reached at and also consider post-hoc power estimation regarding the adequacy of sample size to detect meaningful differences between the two groups.

Line 110-11: “fewer than one percent of our total sample stated that they were not receiving assistance when the survey took place.” So did you exclude such samples from analysis?

Line 133: “at the time of the survey…….. 4% received both”. How did you handle such samples in the analysis?

Line 132-136: “There was no formal process or decision rule for the allocation of beneficiaries to GFD or e-vouchers nor is there written documentation on these allocations.” Have you had any discussion with the program implementers to get better understanding of how the allocations were typically made? Do you anticipate any baseline differences in the nutritional or economic status of the households included into the two modalities?

Line 154-55: Acute malnutrition was only measured using the WHZ index. However, as WHZ is dependent on height, it can potentially be resistant to change, specially in setting where there is active decline in stunting. i.e. progresses made in weight can proportionally be masked by gains in height. I suggest, to include other indices of acute malnutrition (e.g. WAZ, and MUAC) in the analysis, given such information is readily available in your data.

Line 164-73: what was the basis for selecting these variables for adjustment? Statistical? Theoretical/ conceptual framework?

Can you please provide a brief description what you mean by “single difference models”?

Line 179-80: “we include an interaction term between child sex and receipt of an e-voucher to assess whether e-voucher receipt has differential impacts by sex.” Why you the basis for theinteraction assessment? Was that a priori hypothesis?

Line 193-202 and table 1: comparison in basic characteristics should be based on formal statistical tests.

I think duration of support (food ration or e voucher) is a key variable that should be described in Table 1. If this variable is also significantly imbalanced between the two groups on this variable, then it needs to be further adjusted in the multivariable models.

Table 2: what might be behind the R^2 value? Important predictors/confounders missing?

Figure 1 and 2: Please clearly label the x axis.

Discussion

The discussion is superficial and consider the following two comments (1) as clearly described by the reviewers the section does not explain why there was an association between receiving food voucher; (2) the discussion does not provide a strong explanation why electronic food voucher instead of a food ration is associated with improvements in the linear growth but not measures of acute malnutrition.

Line 265-66: the sentence is not clear

2. Please include additional information regarding the survey or questionnaire used in the study and ensure that you have provided sufficient details that others could replicate the analyses. For instance, if you developed a questionnaire as part of this study and it is not under a copyright more restrictive than CC-BY, please include a copy, in both the original language and English, as Supporting Information. Moreover, please include more details on how the questionnaire was pre-tested, and whether it was validated. "

3. Please provide additional details regarding participant consent. As your study included minors, state whether you obtained consent from parents or guardians.

Reviewers' comments:

Reviewer's Responses to Questions

**Comments to the Author**

1. Is the manuscript technically sound, and do the data support the conclusions?

Reviewer #1: Partly

Reviewer #2: Yes

2. Has the statistical analysis been performed appropriately and rigorously? 

Reviewer #1: Yes

Reviewer #2: Yes

3. Have the authors made all data underlying the findings in their manuscript fully available?

Reviewer #1: Yes

Reviewer #2: Yes

4. Is the manuscript presented in an intelligible fashion and written in standard English?

Reviewer #1: Yes

Reviewer #2: Yes

5. Review Comments to the Author

Reviewer #1: This is an interesting topic of current value. The paper is well written although more clarity is needed in places. Overall, this study misses out on any attempt to explain why there was an association. Whilst the authors acknowledge this it is important to know how these interventions are working, especially if the information is to inform future use.

Specific points below

Introduction

Generally, the introduction very short. It would be beneficial if the authors were to refer to some of the more recently published studies of cash and vouchers in humanitarian settings e.g. REFANI studies, Grellety, Concern/Cornell.

Also it would be good for some reference to be made here to other cash/voucher programmes in Bangladesh, if any, with any evidence.

What is the nutritional status of children in this population in this area?

Ln 1 Please check reference #1 is correct as I cannot find the data it is referring too. Reference 1 is a website and not a specific reference.

Ln 57 Could the authors state here that the e-voucher is unconditional (assuming it is).

Setting and methods

Setting and sample

I find the description of the survey could be better. It would be good to understand the objective of the primary survey and whether it was designed also to measure the research question of this paper (intentional study). Or whether this is a sub-analysis on data that was collected anyway. Who was responsible for what? Who did the data collection? Were standardisation methods used in training sessions? What was the sample size and how was it calculated?

Ln 84 please define the type of survey e.g. cross-sectional

Lns 88-89 I do not see the relevance of the following sentence “The survey also included Bangladeshi households living in the host community. However, they do not receive the food assistance discussed in this paper and so are excluded from our analysis.”

Access to food assistance

The authors mention a sample size of 523 children; it would be good to know how this was split between the different interventions

Ln 118 Reference 8 only states the categorisation of household sizes

Ln 118 and 119 “(After our survey was complete, WFP introduced a fourth category for households with more than 11 members [8])” – is this relevant to this study?

Ln 125 Please check if this reference (#8) is correct here

Ln 132-133 “At the time of the survey, across all households, 62%, received GFD, 34% received e-vouchers and 4% received both”. Does this refer to the original sample or the sub-sample? It would be a good idea to make sure information concerning the the original sample and sub-sample are clearly demarcated

How did the authors deal with these 4% of households that received both e-voucher and food?

It would be very interesting to know how long each family had had access to each transfer, whether this was different or not. At the same time was there any information on how the transfers had been used? E.g. Was food sold and the money used for other types of foods or non-food goods? Were e-vouchers exchanged for other (non-food) goods?

Ln 145-146 “Thus, while access to e-vouchers was not randomized and thus our study is associational, there is no evidence to suggest that access to e-vouchers was linked to specific child or household characteristics.” Was this assumption checked?

Ln 145 “Thus, while access to e-vouchers was not randomized and thus our study is associational...” I would argue that this is not the only reason and that non-randomised studies under the right conditions may offer more than association.

Outcome measures

Ln 151 Please check reference 9 is valid as it mentions references for school-aged children and adolescents

Statistical analysis

As mentioned above it would be good to know how the authors handled those households receiving both food and e-voucher

Results

The results are lacking the number of children in either group. Sample sizes would be useful here (as well as in the tables)

It would be good to present the data without stating clarifying information e.g. ‘relatively few’ and ‘few’. Especially as 17% is termed a ‘significant fraction’ and 29% termed ‘relatively few’.

Ln 199 food ratio should be food ration

Lns 199-202 Can this be re-written so not to use ‘less or more likely’ but rather e.g. ‘there were more girls than boys in the households receiving e-vouchers….etc.” The last sentence reads a bit odd and could be added to the preceding sentence

Lns 216-217 Please make clearer by adding that these results are for the whole sample

Lns 237-238 The authors say that “The magnitudes of these associations are small and are not statistically significant”. However, there are some significant differences that could be mentioned

Discussion

Ln 262 The authors write “Rather, receipt of an e-voucher is associated with an increase in HAZ of 0.38SD.” I would caution about inferring a trend here.

Ln 266 I suggest not to use the word ‘impact’ here as it infers causality

Lns 264-268 These two sentences could be clearer as I am not entirely sure what the authors are saying here. “Receipt of the e-voucher is associated with a lower risk of stunting, though this is imprecisely measured once we include a full set of controls. It is possible that this imprecision arises because, as Figure 1 suggests, the impact of e-vouchers, is concentrated on children with very low HAZ scores, well below the -2 cut off used to denote stunting”.

Ln 271 Change ‘on’ to ‘of’

Reviewer #2: Comments to the Author

The present study aimed to examine to examine associations between receipt of an electronic food voucher (e-voucher) compared to food rations on the nutritional status of Rohingya children living in refugee camps in Bangladesh. This is an important topic and data this area is scarce, making this a valuable contribution to the literature. Despite this strength, I have some suggestions for improvement.

Abstract;

- I would recommend to rewrite the abstract for better clarity. For example, the phrase “…but this association is imprecisely measured: & “There is suggestive evidence that this association is larger for girls than boys”

- It is also good to describe the methodology in more detail. I am not sure what the study design the authors used and specific outcomes they are looking for.

- I strongly advise the authors to report results in clearer statements.

Introduction:

- It would also be helpful for the authors to build more of a case for the correlates that they chose to examine here. The introduction need a major revision and shall cover some of the relevant published literature on the issue and need to be explicit enough to show the reader the gap in literature rather than just reporting “little is known about the comparative effects of food and voucher payments on children’s nutritional staus”

- The introduction could better be structured and expanded.

- It would also be helpful for the authors to build a concrete question mainly focusing on specific aspects of child nutrition (macro or micro)?

Methods:

- Some additional detail and structure regarding the methods would be helpful. For example, any theoretical assumptions used to calculate their sample size? What was the anticipated difference in wasting and stunting between the groups? It is also better to be specific in the use of phrases such as chronic undernutrition could be better replaces with stunting and acute malnutrition with wasting etc

- Why would the authors restrict to sub samples given that they are the one who collected the data? Why only some covariates are measured in the subsample? Which covariates? The authors need to describe in details about this issue otherwise the study is liable to selection bias.

- It is better to structure the methods in to sub-sections such as sample size and sampling methods, interventions, measurements, quality control, data management and analysis etc.

- Please indicate and describe the main covariates collected in the study

- More information about measures employed by the authors to keep the quality of data needed.

- The authors need to justify why they included stunting as an outcome measure than other outcome measures that would be relevant in humanitarian settings. This is very important in a way that the main purpose of the support in humanitarian setting is not related to improve child growth rather to overcome acute food shortage.

- Any conceptual framework used to organize the data collection instrument?

Results and Discussions

- Table 1 need to be revised as it is confusing for reader in this format. Please use a table with columns reporting number and frequencies for each covariate reported and disaggregated by the type of intervention, additional column reporting p-value. (please refer STROBE- papers reporting results from either RCT or case control studies)

- I am not sure the relevance of reporting the graphs 1 and 2. I feel that this is a duplication of reporting of results.

- The authors need to clearly report the result obtained from the regression output . Some of the terms or phrases they use may mislead readers, e.g. Line 233 “ .. likelihood that the child is stunted but this association is only marginally statistically significant” this is a misleading statement. It need to be a clear message to the reader that there is no association. (please also revise the abstract section not to misinform readers).

- The authors need to critically discuss on why would the intervention improve the HAZ score and but failed to be reflected in reduction of stunting prevalence? Is there any justification on this?

- The conclusion “ Our results suggest that transitioning from food rations to electronic food vouchers does not adversely affect child nutritional status and may in fact be beneficial” seems over reporting given the limitation in the study methodology. The authors may tone down the conclusion;

6. PLOS authors have the option to publish the peer review history of their article (what does this mean?). If published, this will include your full peer review and any attached files.

Reviewer #1: No

Reviewer #2: Yes: Seifu Hagos Gebreyesus

---

## [Author Response · Author response to Decision Letter 0]

15 Feb 2020

Food transfers, electronic food vouchers and child nutritional status among Rohingya children living in Bangladesh:

Response to comments from editor and reviewers

Editor

Abstract

In the methods sub-section please clearly identify the study design of the study and sample sizes of the two groups being compared.

Response: We have done so. We write, “This is an associational study using cross-sectional data.” Sample sizes have been added.

“The magnitude of the association is large, 0.38 SD” can you please provide the confidence interval for this effect measure?

Response: This has been added.

The sentence “Receipt of an e-voucher is associated with reduced stunting, but this association is imprecisely measured” is not clear.

Response: This sentence has been deleted and replaced with, “Receipt of an e-voucher is not associated with stunting when a full set of control variables are included.”

Introduction

This section looks shallow. Please provide a brief overview of the findings of previous studies on the topic.

Response: We have done so. Reviewer #1’s suggestions were especially helpful and we have incorporated these into our revised introduction. 

Methods

Please describe how the sample size of 523 was reached at and also consider post-hoc power estimation regarding the adequacy of sample size to detect meaningful differences between the two groups.

Response: This section has been extensively re-written to address this concern. We now include the following text: 

“In October and November 2018, we conducted a cross-sectional survey of FDMN in collaboration with the Bangladesh Institute of Development Studies and the non-governmental organization Action Against Hunger. The purpose of the survey was to document the food security and nutrition status of the Rohingya, to understand how they generated income, assess the extent to which they relied on food coping strategies and to review the role played by food assistance efforts provided by international donors in improving food security and nutrition. The survey included both Rohingya who had arrived before August 2017 - the older wave of migrants - and those arriving after the mass expulsion in September and October 2017 - the newer wave of migrants. The Rohingya in the older migration waves were selected through simple random sampling in the Nayapara Registered Camp. The Rohingya in the most recent migration wave were sampled using two-stage clustered random sampling, using the blocks (sub-camps) within each camp (strata) as the clusters. The final sample consists of 1,308 Rohingya households from the newer wave, 781 Rohingya households from the older waves. The survey instrument covered household demographics, subjective wellbeing, consumption and expenditure, coping strategies, nutrition outcomes, livelihoods and income profiles, and access to assistance. These data were collected by enumerators trained and employed by the Bangladesh Institute of Development Studies and Action Against Hunger. Collection of the anthropometric data drew on the Standardized Monitoring and Assessment of Relief and Transitions (SMART) methodology used by international organizations and humanitarian practitioners. SMART includes methods for standardizing anthropometric measurements as well as software to check data quality and flag problems.”

“For this paper, we undertook a sub-analysis of these data. Specifically, we focus on those children in our sampled households who were aged between 6 and 23 months at the time of the survey. Children in this age range are within the “1000 days window” critical for child nutrition. They are no longer being exclusively breastfed; for children in this age range, the availability of adequate quantities of a range of nutritious foods is critical for their health and development. In our sample, there are 523 children in this age range. 362 children live in households receiving the food ration; 161 children live in households receiving the e-voucher. An ex-post power calculation showed that this sample was large enough to detect a: 0.35SD difference in height-for-age z score between children living in households receiving food rations and e-vouchers; and a 0.25SD difference in weight-for-height z scores between children living in households receiving food rations and e-vouchers.”

Line 110-11: “fewer than one percent of our total sample stated that they were not receiving assistance when the survey took place.” So did you exclude such samples from analysis?

Response: Households that did not receive assistance were excluded from our analysis. This has been noted.

Line 133: “at the time of the survey…….. 4% received both”. How did you handle such samples in the analysis? 

Response: Households that stated that they received both forms of assistance were excluded from our analysis. This has been noted.

Line 132-136: “There was no formal process or decision rule for the allocation of beneficiaries to GFD or e-vouchers nor is there written documentation on these allocations.” Have you had any discussion with the program implementers to get better understanding of how the allocations were typically made? Do you anticipate any baseline differences in the nutritional or economic status of the households included into the two modalities?

Response: Yes, we did meet with program implementers to understand how these allocations were made and we have incorporated a summary of these discussions into our revised paper. We write, “As part of our research for this study, in May 2019 we met the Emergency Coordinator for the World Food Programme Rohingya Refugee Response and his staff. He indicated that these proportions reflected the amount of cash and of food that WFP had available to distribute. He noted that there was no formal process or decision rule for the allocation of beneficiaries to GFD or e-vouchers nor was there written documentation on these allocations. He and his staff described the process of selecting areas where e-vouchers would be used in the following way. The use of e-vouchers required new shops to be constructed and shops require land. WFP was not permitted to purchase this land. Instead, the camps were demarcated into five catchment areas and within these, the Government of Bangladesh provided land (where available) at certain places within each for shops to be located. There was no formal process or criteria for the decision to provide land (and build a shop) in any given place and WFP had no choice regarding where these shops were sited; maps provided to us by WFP show that most of these shops are co-located in places where the GFD is given out. Once this decision was made, a certain number of households in the proximity of the shop were enrolled into e-vouchers. Beneficiaries themselves had no choice as to whether they would receive GFD or e-vouchers. Access to e-vouchers was not randomized and thus our study is associational. However, WFP staff indicated that access to e-vouchers was not linked to specific child or household characteristics.”

That said, just because access to e-vouchers was not linked by program implementers to specific child or household characteristics does not mean that there are no differences in economic status of households included into the two modalities. We test for such differences in our revised Table 1 and we include child, maternal and household characteristics as control variables in our regressions.

Line 154-55: Acute malnutrition was only measured using the WHZ index. However, as WHZ is dependent on height, it can potentially be resistant to change, specially in setting where there is active decline in stunting. i.e. progresses made in weight can proportionally be masked by gains in height. I suggest, to include other indices of acute malnutrition (e.g. WAZ, and MUAC) in the analysis, given such information is readily available in your data.

Response: We now include these. We do not find a statistically significant association between these outcomes and receipt of the e-voucher.

Line 164-73: what was the basis for selecting these variables for adjustment? Statistical? Theoretical/ conceptual framework?

Response: We drew on the UNICEF conceptual framework for the selection of these variables. This has now been noted in the paper.

Can you please provide a brief description what you mean by “single difference models”?

Response: We have dropped this phrase.

Line 179-80: “we include an interaction term between child sex and receipt of an e-voucher to assess whether e-voucher receipt has differential impacts by sex.” Why you the basis for the interaction assessment? Was that a priori hypothesis?

Response: There is a large literature on gender bias in nutrition-related outcomes in south Asia. In light of this, we wondered if receipt of e-vouchers had gender differentiated associations with the anthropometric outcomes we consider in our paper.

Line 193-202 and table 1: comparison in basic characteristics should be based on formal statistical tests.

I think duration of support (food ration or e voucher) is a key variable that should be described in Table 1. If this variable is also significantly imbalanced between the two groups on this variable, then it needs to be further adjusted in the multivariable models.

Response: Formal statistical tests are now reported in Table 1. We know frequency of transfer (which we adjust for in our multivariable models) but we do not have data on duration of support.

Table 2: what might be behind the R^2 value? Important predictors/confounders missing?

Response: We do not have measurements of maternal height. Maternal height is associated with HAZ and so, its absence will result in lower R^2.

Figure 1 and 2: Please clearly label the x axis.

Response: We have done so.

Discussion

The discussion is superficial and consider the following two comments (1) as clearly described by the reviewers the section does not explain why there was an association between receiving food voucher; (2) the discussion does not provide a strong explanation why electronic food voucher instead of a food ration is associated with improvements in the linear growth but not measures of acute malnutrition.

Response: We have added new material that addresses, to the best of our ability (given our data limitations) these points. We write:

“Our study has only limited information on why these positive associations exist. We note that women in male headed households receiving the e-voucher were 11 percentage points more likely to decide solely or jointly how to use the food assistance that their households had received compared to women in male headed households receiving the GFD. Households receiving the e-voucher reported being able to make the assistance they received last until the next payment [17]. We also note that there is one published study in a developing country setting, using a randomized control trial design, where treatment arms included a food transfer and a food voucher. This study, fielded in Ecuador, found that both treatments increased the quantity and quality of food consumed by recipient households. However, food transfers led to relatively larger increases in calories consumed while food vouchers led to relatively larger increases in household dietary diversity [18]. Consistent with the Ecuador study, households in our study who received an e-voucher consumed a larger number of food groups than households that received the GFD. If households receiving the e-voucher purchased a wider range of foods then children in this age group may have consumed a more diverse diet, including animal source foods that Semba et al [19] suggest are linked to more rapid linear growth note. We have data on household level consumption but not data on child food consumption; consequently, we are unable to fully investigate this hypothesis. Improved dietary diversity, increased women’s control over the transfer and the ability of households to making the transfer last until the next payment are all associated with receipt of the e-voucher. All may have played some role in improving HAZ; our data, however, do not allow us to disentangle their possible effects. We also note that hygiene conditions in these settlements are poor [9] and that poor sanitation is associated with poorer WHZ. Neither e-vouchers or the GFD directly affect hygiene conditions and this may explain why there is no association between e-vouchers and WHZ.”

Line 265-66: the sentence is not clear

Response: This sentence has been deleted.

http://www.journals.plos.org/plosone/s/file?id=wjVg/PLOSOne_formatting_sample_main_body.pdf and http://www.journals.plos.org/plosone/s/file?id=ba62/PLOSOne_formatting_sample_title_authors_affiliations.pdf.

Response: Noted.

2. Please include additional information regarding the survey or questionnaire used in the study and ensure that you have provided sufficient details that others could replicate the analyses. For instance, if you developed a questionnaire as part of this study and it is not under a copyright more restrictive than CC-BY, please include a copy, in both the original language and English, as Supporting Information. Moreover, please include more details on how the questionnaire was pre-tested, and whether it was validated. "

Response. It built on previous survey instruments fielded in these localities; modules not previously implemented were pre-tested before the questionnaire was fielded. The survey instrument has been uploaded under supporting information.

 3. Please provide additional details regarding participant consent. As your study included minors, state whether you obtained consent from parents or guardians.

Response. This statement has been added.

Response: Noted.

 

Reviewer #1

This is an interesting topic of current value. The paper is well written although more clarity is needed in places. Overall, this study misses out on any attempt to explain why there was an association. Whilst the authors acknowledge this it is important to know how these interventions are working, especially if the information is to inform future use. Specific points below.

Response: Thank you for these helpful comments. We hope we have satisfactorily addressed the concerns you have raised.

Introduction

Generally, the introduction very short. It would be beneficial if the authors were to refer to some of the more recently published studies of cash and vouchers in humanitarian settings e.g. REFANI studies, Grellety, Concern/Cornell. Also it would be good for some reference to be made here to other cash/voucher programmes in Bangladesh, if any, with any evidence.

What is the nutritional status of children in this population in this area?

Response: Thank you for these excellent suggestions. We have included the studies you suggested, added a reference to relevant work done elsewhere in Bangladesh and have noted the nutritional status of children in this population.

Ln 1 Please check reference #1 is correct as I cannot find the data it is referring too. Reference 1 is a website and not a specific reference.

Response: Apologies, the numbers quoted are correct but the reference was incorrect. This has been fixed. 

Ln 57 Could the authors state here that the e-voucher is unconditional (assuming it is).

Response: They were unconditional and we have stated so here. 

Setting and methods

Setting and sample

I find the description of the survey could be better. It would be good to understand the objective of the primary survey and whether it was designed also to measure the research question of this paper (intentional study). Or whether this is a sub-analysis on data that was collected anyway. Who was responsible for what? Who did the data collection? Were standardisation methods used in training sessions? What was the sample size and how was it calculated?

Response: This section has been extensively re-written to address this concern. We now include the following text: 

“In October and November 2018, we conducted a cross-sectional survey of FDMN in collaboration with the Bangladesh Institute of Development Studies and the non-governmental organization Action Against Hunger. The purpose of the survey was to document the food security and nutrition status of the Rohingya, to understand how they generated income, assess the extent to which they relied on food coping strategies and to review the role played by food assistance efforts provided by international donors in improving food security and nutrition. The survey included both Rohingya who had arrived before August 2017 - the older wave of migrants - and those arriving after the mass expulsion in September and October 2017 - the newer wave of migrants. The Rohingya in the older migration waves were selected through simple random sampling in the Nayapara Registered Camp. The Rohingya in the most recent migration wave were sampled using two-stage clustered random sampling, using the blocks (sub-camps) within each camp (strata) as the clusters. The final sample consists of 1,308 Rohingya households from the newer wave, 781 Rohingya households from the older waves. The survey instrument covered household demographics, subjective wellbeing, consumption and expenditure, coping strategies, nutrition outcomes, livelihoods and income profiles, and access to assistance. These data were collected by enumerators trained and employed by the Bangladesh Institute of Development Studies and Action Against Hunger. Collection of the anthropometric data drew on the Standardized Monitoring and Assessment of Relief and Transitions (SMART) methodology used by international organizations and humanitarian practitioners. SMART includes methods for standardizing anthropometric measurements as well as software to check data quality and flag problems.”

“For this paper, we undertook a sub-analysis of these data. Specifically, we focus on those children in our sampled households who were aged between 6 and 23 months at the time of the survey. Children in this age range are within the “1000 days window” critical for child nutrition. They are no longer being exclusively breastfed; for children in this age range, the availability of adequate quantities of a range of nutritious foods is critical for their health and development. In our sample, there are 523 children in this age range. 362 children live in households receiving the food ration; 161 children live in households receiving the e-voucher. An ex-post power calculation showed that this sample was large enough to detect a: 0.35SD difference in height-for-age z score between children living in households receiving food rations and e-vouchers; and a 0.25SD difference in weight-for-height z scores between children living in households receiving food rations and e-vouchers.”

Ln 84 please define the type of survey e.g. cross-sectional

Response: This has been added. 

Lns 88-89 I do not see the relevance of the following sentence “The survey also included Bangladeshi households living in the host community. However, they do not receive the food assistance discussed in this paper and so are excluded from our analysis.”

Response: These sentences have been deleted.

Access to food assistance

The authors mention a sample size of 523 children; it would be good to know how this was split between the different interventions.

Response: This has been added.

Ln 118 Reference 8 only states the categorisation of household sizes.

Response: Reference 7 gives information on how distributions vary for food rations by household size. Reference 8 has been deleted.

Ln 118 and 119 “(After our survey was complete, WFP introduced a fourth category for households with more than 11 members [8])” – is this relevant to this study? 

Response: This sentence has been deleted.

Ln 125 Please check if this reference (#8) is correct here.

Response: We have deleted this reference and replaced it with another that states this more clearly.

Ln 132-133 “At the time of the survey, across all households, 62%, received GFD, 34% received e-vouchers and 4% received both”. Does this refer to the original sample or the sub-sample? It would be a good idea to make sure information concerning the the original sample and sub-sample are clearly demarcated. 

Response: This refers to all households interviewed (ie the 1,308 Rohingya households from the newer wave of refugees and 781 Rohingya from the older waves of refugees). This has been clarified in the text.

How did the authors deal with these 4% of households that received both e-voucher and food?

Response: Households that stated that they received both forms of assistance were excluded from our analysis. This has been noted.

It would be very interesting to know how long each family had had access to each transfer, whether this was different or not. At the same time was there any information on how the transfers had been used? E.g. Was food sold and the money used for other types of foods or non-food goods? Were e-vouchers exchanged for other (non-food) goods?

Response: We do not know how long each family had had access to each transfer. Re-sale of the GFD is relatively uncommon. 76% of households that received some WFP assistance reported not selling it. Only 15% said they sold it for food aid and 12% exchanged or bartered it for other commodities. This is now reported in the paper.

Ln 145-146 “Thus, while access to e-vouchers was not randomized and thus our study is associational, there is no evidence to suggest that access to e-vouchers was linked to specific child or household characteristics.” Was this assumption checked?

Ln 145 “Thus, while access to e-vouchers was not randomized and thus our study is associational...” I would argue that this is not the only reason and that non-randomised studies under the right conditions may offer more than association.

Response: We have clarified the process by which households were allocated to food transfers or e-vouchers. We write, “As part of our research for this study, in May 2019 we met the Emergency Coordinator for the World Food Programme Rohingya Refugee Response and his staff. He indicated that these proportions reflected the amount of cash and of food that WFP had available to distribute. He noted that there was no formal process or decision rule for the allocation of beneficiaries to GFD or e-vouchers nor was there written documentation on these allocations. He and his staff described the process of selecting areas where e-vouchers would be used in the following way. The use of e-vouchers required new shops to be constructed and shops require land. WFP was not permitted to purchase this land. Instead, the camps were demarcated into five catchment areas and within these, the Government of Bangladesh provided land (where available) at certain places within each for shops to be located. There was no formal process or criteria for the decision to provide land (and build a shop) in any given place and WFP had no choice regarding where these shops were sited; maps provided to us by WFP show that most of these shops are co-located in places where the GFD is given out. Once this decision was made, a certain number of households in the proximity of the shop were enrolled into e-vouchers. Beneficiaries themselves had no choice as to whether they would receive GFD or e-vouchers. Access to e-vouchers was not randomized and thus our study is associational. However, WFP staff indicated that access to e-vouchers was not linked to specific child or household characteristics.”

Outcome measures

Ln 151 Please check reference 9 is valid as it mentions references for school-aged children and adolescents.

Response: Reference 10 contains all the information discussed here and so we have deleted reference 9 as it is redundant.

Statistical analysis

As mentioned above it would be good to know how the authors handled those households receiving both food and e-voucher.

Response: Households that stated that they received both forms of assistance were excluded from our analysis. This has been noted.

Results

The results are lacking the number of children in either group. Sample sizes would be useful here (as well as in the tables).

Response: As requested above, we added these numbers to the section “Setting and sample”. They have also been added to all tables.

It would be good to present the data without stating clarifying information e.g. ‘relatively few’ and ‘few’. Especially as 17% is termed a ‘significant fraction’ and 29% termed ‘relatively few’.

Response: These clarifying statements have been deleted.

Ln 199 food ratio should be food ration 

Response: Corrected.

Lns 199-202 Can this be re-written so not to use ‘less or more likely’ but rather e.g. ‘there were more girls than boys in the households receiving e-vouchers….etc.” The last sentence reads a bit odd and could be added to the preceding sentence

Response: These sentences have been re-written.

Lns 216-217 Please make clearer by adding that these results are for the whole sample.

Response: Clarified.

Lns 237-238 The authors say that “The magnitudes of these associations are small and are not statistically significant”. However, there are some significant differences that could be mentioned.

Response: This has been re-written, clarifying that we are referring to the associations between receipt of e-vouchers and WHZ and receipt of e-vouchers and wasting.

Discussion

Ln 262 The authors write “Rather, receipt of an e-voucher is associated with an increase in HAZ of 0.38SD.” I would caution about inferring a trend here.

Response: re-written to make clear that we are not inferring a trend.

Ln 266 I suggest not to use the word ‘impact’ here as it infers causality.

Response: The word impact has been deleted and the sentence re-written.

Lns 264-268 These two sentences could be clearer as I am not entirely sure what the authors are saying here. “Receipt of the e-voucher is associated with a lower risk of stunting, though this is imprecisely measured once we include a full set of controls. It is possible that this imprecision arises because, as Figure 1 suggests, the impact of e-vouchers, is concentrated on children with very low HAZ scores, well below the -2 cut off used to denote stunting”.

Response: We have re-written these sentences to clarify that a possible reason why we observe an impact on HAZ but not on stunting is because the impact on HAZ is concentrated on children with HAZ well below the -2 cut-off used to denote that a child is stunted.

Ln 271 Change ‘on’ to ‘of’ 

Response: Corrected.

 

Reviewer #2: Comments to the Author

The present study aimed to examine to examine associations between receipt of an electronic food voucher (e-voucher) compared to food rations on the nutritional status of Rohingya children living in refugee camps in Bangladesh. This is an important topic and data this area is scarce, making this a valuable contribution to the literature. Despite this strength, I have some suggestions for improvement.

Response: Thank you for these helpful comments. We hope we have satisfactorily addressed the concerns you have raised.

- I would recommend to rewrite the abstract for better clarity. For example, the phrase “…but this association is imprecisely measured: & “There is suggestive evidence that this association is larger for girls than boys”

Response: We have done so. We now write, “Receipt of an e-voucher is not associated with stunting when a full set of control variables are included.” We also write, “We cannot reject the null hypothesis that these associations differ by child sex.”

- It is also good to describe the methodology in more detail. I am not sure what the study design the authors used and specific outcomes they are looking for.

Response: We have added details on methodologu, study design and outcomes.

- I strongly advise the authors to report results in clearer statements.

Response: As noted above, we have revised the abstract, using clearer statements to when reporting our results.

Introduction:

- It would also be helpful for the authors to build more of a case for the correlates that they chose to examine here. The introduction need a major revision and shall cover some of the relevant published literature on the issue and need to be explicit enough to show the reader the gap in literature rather than just reporting “little is known about the comparative effects of food and voucher payments on children’s nutritional staus”

- The introduction could better be structured and expanded.

- It would also be helpful for the authors to build a concrete question mainly focusing on specific aspects of child nutrition (macro or micro)?

Response: Thank you for these excellent suggestions. We have included a review of the relevant published literature, have expanded the introduction and clarified which dimensions of child nutrition that we study. 

Methods:

- Some additional detail and structure regarding the methods would be helpful. For example, any theoretical assumptions used to calculate their sample size? What was the anticipated difference in wasting and stunting between the groups? It is also better to be specific in the use of phrases such as chronic undernutrition could be better replaces with stunting and acute malnutrition with wasting etc

- Why would the authors restrict to sub samples given that they are the one who collected the data? Why only some covariates are measured in the subsample? Which covariates? The authors need to describe in details about this issue otherwise the study is liable to selection bias.

- More information about measures employed by the authors to keep the quality of data needed.

- Any conceptual framework used to organize the data collection instrument?

Response: We take these four points together. We note that the section on “setting and sample” has been extensively re-written to address your concern. We now include the following text: 

“In October and November 2018, we conducted a cross-sectional survey of FDMN in collaboration with the Bangladesh Institute of Development Studies and the non-governmental organization Action Against Hunger. The purpose of the survey was to document the food security and nutrition status of the Rohingya, to understand how they generated income, assess the extent to which they relied on food coping strategies and to review the role played by food assistance efforts provided by international donors in improving food security and nutrition. The survey included both Rohingya who had arrived before August 2017 - the older wave of migrants - and those arriving after the mass expulsion in September and October 2017 - the newer wave of migrants. The Rohingya in the older migration waves were selected through simple random sampling in the Nayapara Registered Camp. The Rohingya in the most recent migration wave were sampled using two-stage clustered random sampling, using the blocks (sub-camps) within each camp (strata) as the clusters. The final sample consists of 1,308 Rohingya households from the newer wave, 781 Rohingya households from the older waves. The survey instrument covered household demographics, subjective wellbeing, consumption and expenditure, coping strategies, nutrition outcomes, livelihoods and income profiles, and access to assistance. These data were collected by enumerators trained and employed by the Bangladesh Institute of Development Studies and Action Against Hunger. Collection of the anthropometric data drew on the Standardized Monitoring and Assessment of Relief and Transitions (SMART) methodology used by international organizations and humanitarian practitioners. SMART includes methods for standardizing anthropometric measurements as well as software to check data quality and flag problems.”

“For this paper, we undertook a sub-analysis of these data. Specifically, we focus on those children in our sampled households who were aged between 6 and 23 months at the time of the survey. Children in this age range are within the “1000 days window” critical for child nutrition. They are no longer being exclusively breastfed; for children in this age range, the availability of adequate quantities of a range of nutritious foods is critical for their health and development. In our sample, there are 523 children in this age range. 362 children live in households receiving the food ration; 161 children live in households receiving the e-voucher. An ex-post power calculation showed that this sample was large enough to detect a: 0.35SD difference in height-for-age z score between children living in households receiving food rations and e-vouchers; and a 0.25SD difference in weight-for-height z scores between children living in households receiving food rations and e-vouchers.”

- It is better to structure the methods in to sub-sections such as sample size and sampling methods, interventions, measurements, quality control, data management and analysis etc.

Response: Both the editor and the other referee were happy with the way in which the methods section is structured and for this reason, we have not re-structured this material. However, if there are specific points that you require more information about, please let us know and we will edit this section accordingly.

- Please indicate and describe the main covariates collected in the study

Response: the covariates used for this study are described in the section “statistical analysis”.

- The authors need to justify why they included stunting as an outcome measure than other outcome measures that would be relevant in humanitarian settings. This is very important in a way that the main purpose of the support in humanitarian setting is not related to improve child growth rather to overcome acute food shortage.

Response: We now note the following. “Studies of the impact of cash or food interventions in humanitarian settings focus on short-term measures of nutritional status such as weight-for-height and acute undernutrition, wasting. But because the Rohingya are expected to remain in these refugee camps for a considerable length of time, it is also of interest to see how these different transfer modalities affect child growth.”

Results and Discussions

- Table 1 need to be revised as it is confusing for reader in this format. Please use a table with columns reporting number and frequencies for each covariate reported and disaggregated by the type of intervention, additional column reporting p-value. (please refer STROBE- papers reporting results from either RCT or case control studies).

Response: We have made these requested changes.

- I am not sure the relevance of reporting the graphs 1 and 2. I feel that this is a duplication of reporting of results.

Response: These graphs are useful because, as we discuss in the Results section, a possible reason why we observe an impact on HAZ but not on stunting is because the impact on HAZ is concentrated on children with HAZ well below the -2 cut-off used to denote that a child is stunted (see Figure 1). Hence, we would prefer to retain these graphs.

- The authors need to clearly report the result obtained from the regression output . Some of the terms or phrases they use may mislead readers, e.g. Line 233 “ .. likelihood that the child is stunted but this association is only marginally statistically significant” this is a misleading statement. It need to be a clear message to the reader that there is no association. (please also revise the abstract section not to misinform readers).

Response: We have re-written both this text and the abstract to make clear that there is no association between receiving an e-voucher and stunting.

- The authors need to critically discuss on why would the intervention improve the HAZ score and but failed to be reflected in reduction of stunting prevalence? Is there any justification on this?

Response: A possible reason why we observe an impact on HAZ but not on stunting is because the impact on HAZ is concentrated on children with HAZ well below the -2 cut-off used to denote that a child is stunted (see Figure 1). This is noted in the text.

- The conclusion “ Our results suggest that transitioning from food rations to electronic food vouchers does not adversely affect child nutritional status and may in fact be beneficial” seems over reporting given the limitation in the study methodology. The authors may tone down the conclusion;

Response: Agreed. This has been toned down.

---

## [Editor Report · Decision Letter 1]

2 Mar 2020

Food transfers, electronic food vouchers and child nutritional status among Rohingya children living in Bangladesh

PONE-D-19-25181R1

Dear Dr. Hoddinott,

We are pleased to inform you that your manuscript has been judged scientifically suitable for publication and will be formally accepted for publication once it complies with all outstanding technical requirements.

With kind regards,

Samson Gebremedhin, PhD

Academic Editor

PLOS ONE
---

## [Editor Report · Acceptance letter]

7 Apr 2020

PONE-D-19-25181R1 

Food transfers, electronic food vouchers and child nutritional status among Rohingya children living in Bangladesh 

Dear Dr. Hoddinott:

I am pleased to inform you that your manuscript has been deemed suitable for publication in PLOS ONE. Congratulations! Your manuscript is now with our production department. 

With kind regards,

on behalf of

Dr. Samson Gebremedhin 

Academic Editor

PLOS ONE